# Toxicity, Mitigation, and Chemical Analysis of Aflatoxins and Other Toxic Metabolites Produced by Aspergillus: A Comprehensive Review

**DOI:** 10.3390/toxins17070331

**Published:** 2025-06-30

**Authors:** Habtamu Fekadu Gemede

**Affiliations:** Department of Food Technology and Process Engineering, Wollega University, Nekemte P.O. Box 395, Ethiopia; fekadu_habtamu@yahoo.com

**Keywords:** aflatoxins, *Aspergillus*, food safety, mycotoxins, public health, mitigation strategies, detection methods

## Abstract

Aflatoxins, toxic secondary metabolites produced primarily by *Aspergillus flavus* and *Aspergillus parasiticus*, pose significant risks to food safety, public health, and global trade. These mycotoxins contaminate staple crops such as maize and peanuts, particularly in warm and humid regions, leading to economic losses and severe health effects, including hepatocellular carcinoma, immune suppression, and growth impairment. In addition to aflatoxins, Aspergillus species produce other toxic metabolites such as ochratoxin A, sterigmatocystin, and cyclopiazonic acid, which are associated with nephrotoxic, carcinogenic, and neurotoxic effects, respectively. This review provides a comprehensive analysis of aflatoxin toxicity, mitigation strategies, and chemical detection methods. The toxicity of aflatoxins is discussed in relation to their biochemical mechanisms, carcinogenicity, and synergistic effects with other mycotoxins. Various mitigation approaches, including pre-harvest biocontrol, post-harvest storage management, and novel detoxification methods such as enzymatic degradation and nanotechnology-based interventions, are evaluated. Furthermore, advances in aflatoxin detection, including chromatographic, immunoassay, and biosensor-based methods, are explored to improve regulatory compliance and food safety monitoring. This review underscores the need for integrated management strategies and global collaboration to reduce aflatoxin contamination and its associated health and economic burdens. Future research directions should focus on genetic engineering for resistant crop varieties, climate adaptation strategies, and improved risk assessment models.

## 1. Introduction

Aspergillus species, particularly *Aspergillus flavus* and *Aspergillus parasiticus* (Figure 1), are ubiquitous fungi that thrive in warm and humid environments [1]. These species are notorious for producing aflatoxins, a group of highly toxic and carcinogenic secondary metabolites. Aflatoxins primarily contaminate staple crops such as maize, peanuts, and tree nuts, posing significant risks to food safety and security. The global prevalence of aflatoxin contamination is a major concern, especially in developing countries where agricultural practices and storage conditions are often suboptimal [2]. The production of aflatoxins by Aspergillus species is influenced by environmental factors such as temperature, humidity, and substrate availability. Optimal conditions for aflatoxin production include temperatures between 25 °C and 35 °C, and relative humidity above 80%. These conditions are commonly found in tropical and subtropical regions, where aflatoxin contamination is most prevalent. However, climate change is expanding the geographic range of Aspergillus species, increasing the risk of contamination in previously unaffected areas [3].

Aspergillus species are opportunistic pathogens that infect crops both pre- and post-harvest. Pre-harvest infection occurs when crops are exposed to fungal spores in the field, while post-harvest contamination results from improper storage conditions. Both scenarios contribute to the widespread presence of aflatoxins in the food supply chain, making them a global public health concern [4].

The chemical structure of aflatoxins consists of a difuranocoumarin backbone, which is responsible for their toxicity. Among the various aflatoxins, aflatoxin B1 (AFB1) is the most toxic and prevalent [5]. The presence of multiple aflatoxins in food and feed complicates the challenge of ensuring food safety [6]. The global burden of aflatoxin contamination is exacerbated by the lack of effective regulatory frameworks in many developing countries. Weak enforcement of food safety standards, coupled with limited resources for monitoring and control, allows contaminated crops to enter the food supply. This highlights the need for international collaboration and capacity building to address the issue [7]. Public awareness campaigns and education programs can also play a key role in improving food safety practices [8].

The economic impact of aflatoxin contamination is substantial, with billions of dollars lost annually due to crop spoilage, trade restrictions, and health-related costs. In developing countries, where regulatory frameworks are often weak, the public health burden is particularly severe (Table 1). Chronic exposure to aflatoxins is linked to liver cancer, immune suppression, and growth impairment in children, while acute exposure can lead to aflatoxicosis, a potentially fatal condition [9]. Aflatoxin contamination also disrupts international trade, as many countries have strict regulations on permissible aflatoxin levels in imported food and feed. Crops exceeding these limits are often rejected, leading to significant economic losses for exporting nations. For example, African countries lose an estimated $670 million annually due to aflatoxin-related trade restrictions. This economic burden exacerbates poverty and food insecurity in regions already struggling with resource limitations [10].

The public health impact of aflatoxins is equally alarming. Aflatoxin B1, the most toxic and prevalent aflatoxin, is classified as a Group 1 carcinogen by the International Agency for Research on Cancer (IARC). Chronic exposure to even low levels of aflatoxins is a major risk factor for hepatocellular carcinoma (HCC), one of the most common and deadly cancers worldwide. In sub-Saharan Africa and Southeast Asia, where aflatoxin contamination is widespread, HCC incidence rates are among the highest globally [11]. Children are particularly vulnerable to the effects of aflatoxin exposure. Studies have shown that chronic ingestion of aflatoxin-contaminated food is associated with stunted growth, underweightedness, and developmental delays. These effects are thought to result from a combination of immune suppression, reduced nutrient absorption, and direct toxicity to developing tissues. Addressing aflatoxin contamination is critical for improving child health and development, particularly in low-resource settings [12].

In addition to human health, aflatoxins also pose significant risks to livestock and poultry. Contaminated feed reduces animal productivity, causing liver damage, reduced growth rates, and increased susceptibility to diseases. High levels of aflatoxin contamination can lead to acute toxicity and mortality, resulting in significant economic losses for farmers. This, in turn, affects food availability and affordability, further exacerbating food insecurity [13]. The global burden of aflatoxin-related diseases is substantial, particularly in developing countries where food safety regulations are often inadequate. Aflatoxin exposure contributes to millions of cases of liver cancer and other health conditions annually, highlighting the need for effective prevention and control measures. Public awareness campaigns and education programs are essential for reducing aflatoxin exposure and improving food safety [14].

Aflatoxin contamination is most prevalent in tropical and subtropical regions, where climatic conditions favor the growth of Aspergillus species (Table 2). High temperatures and humidity, combined with poor storage practices, create ideal conditions for fungal growth and aflatoxin production. Countries in sub-Saharan Africa, Southeast Asia, and Latin America are particularly affected, with maize and peanuts being the most commonly contaminated crops [15]. However, climate change and global trade have expanded the geographic distribution of aflatoxin contamination, making it a global issue. Rising temperatures and changing precipitation patterns are expected to increase the risk of aflatoxin contamination in previously unaffected regions. For example, recent studies have reported aflatoxin contamination in European countries, where it was previously rare, highlighting the need for global surveillance and mitigation efforts [16].

Surveillance data indicate that aflatoxin levels often exceed regulatory limits in many countries, particularly in developing regions. For instance, a study conducted in Kenya found that 55% of maize samples exceeded the permissible aflatoxin limit of 10 µg/kg. Similarly, high levels of aflatoxin contamination have been reported in peanuts from India and Argentina, underscoring the global nature of the problem [17]. The prevalence of aflatoxin contamination varies depending on agricultural practices, storage conditions, and regulatory enforcement. In developed countries, stringent regulations and advanced storage facilities help reduce contamination levels. However, in developing countries, where resources are limited, aflatoxin contamination remains a significant challenge. Addressing this disparity requires global collaboration and capacity building to improve food safety standards worldwide [18].

## 2. Toxicity of Aflatoxins and Other Toxic Metabolites

### 2.1. Chemical Structure and Classification of Aflatoxins

Aflatoxins are a group of polyketide-derived mycotoxins, with aflatoxin B1 (AFB1) being the most toxic and prevalent. AFB1 is found in milk and dairy products, posing risks to infants and young children [19]. The chemical structure of aflatoxins consists of a difuranocoumarin backbone, which is responsible for their toxicity. The presence of a double bond in the furan ring of AFB1 enhances its reactivity, making it more carcinogenic than other aflatoxins. This structural feature is critical for understanding the mechanisms of aflatoxin toxicity and developing targeted mitigation strategies [20].

Aflatoxins are classified into four major types, B1, B2, G1, and G2, based on their fluorescence under UV light (Figure 2). AFB1 and AFB2 emit blue fluorescence, while AFG1 and AFG2 emit green fluorescence. This classification is useful for analytical purposes, as it allows for the identification and quantification of different aflatoxins in food and feed samples [21].

In addition to the major aflatoxins, Aspergillus species produce other toxic metabolites, such as ochratoxin A, sterigmatocystin, and cyclopiazonic acid. These compounds often co-occur with aflatoxins, complicating the health risks associated with fungal contamination. Understanding the chemical diversity of Aspergillus metabolites is essential for assessing their combined toxicity and developing effective mitigation strategies [22]. The production of aflatoxins is influenced by environmental factors such as temperature, humidity, and substrate availability. Optimal conditions for aflatoxin production include temperatures between 25 °C and 35 °C and relative humidity above 80%. These conditions are commonly found in tropical and subtropical regions, where aflatoxin contamination is most prevalent [23]. Efforts to mitigate aflatoxin contamination must consider the chemical properties of aflatoxins and their interaction with food matrices. For example, aflatoxins are relatively stable under normal cooking temperatures, making them difficult to eliminate through conventional food processing methods. This highlights the need for innovative approaches, such as enzymatic degradation and nanotechnology-based detoxification [24].

### 2.2. Mechanisms of Toxicity

The primary mechanism of aflatoxin toxicity involves the formation of DNA adducts, particularly AFB1-N7-guanine, which leads to mutations in the p53 tumor suppressor gene. This process is a key driver of hepatocellular carcinoma (HCC), one of the most common and deadly cancers worldwide (Figure 3). The reactive epoxide metabolite of AFB1 binds to DNA, causing structural changes that impair DNA replication and repair [25]. In addition to DNA adduct formation, aflatoxins induce oxidative stress by generating reactive oxygen species (ROS). ROS cause lipid peroxidation, protein oxidation, and DNA damage, leading to cellular dysfunction and apoptosis. The liver, being the primary site of aflatoxin metabolism, is particularly vulnerable to oxidative stress, which exacerbates the hepatotoxic effects of aflatoxins [26].

Aflatoxins also impair immune function by suppressing the production of cytokines and antibodies (Figure 4). This immunosuppressive effect increases susceptibility to infections, particularly in populations with a high prevalence of infectious diseases such as HIV/AIDS and malaria. Studies have shown that aflatoxin exposure is associated with increased viral load and disease progression in HIV-positive individuals [27]. The toxicity of aflatoxins is influenced by genetic factors, such as polymorphisms in genes involved in aflatoxin metabolism and detoxification. For example, individuals with certain variants of the glutathione S-transferase (GST) gene are more susceptible to aflatoxin-induced liver damage. Understanding these genetic factors is essential for identifying at-risk populations and developing personalized prevention strategies [28].

In addition to their direct toxic effects, aflatoxins can interact with other mycotoxins, such as fumonisins and ochratoxin A, to produce synergistic effects. These interactions amplify the overall toxicity of contaminated food and feed, complicating risk assessment and mitigation efforts. For example, the combined exposure to aflatoxins and fumonisins has been shown to increase the risk of liver and esophageal cancer [29].

### 2.3. Acute and Chronic Health Effects

Acute aflatoxin exposure can cause aflatoxicosis, a potentially fatal condition characterized by vomiting, abdominal pain, and liver failure. Outbreaks of aflatoxicosis have been reported in several countries (Figure 5), particularly in regions with high levels of aflatoxin contamination in staple foods. For example, a 2004 outbreak in Kenya resulted in over 100 deaths, highlighting the severity of acute aflatoxin toxicity [30]. Chronic exposure to aflatoxins, even at low levels, is associated with a range of health effects, including liver cancer, immune suppression, and growth impairment. Aflatoxin B1 is classified as a Group 1 carcinogen by the International Agency for Research on Cancer (IARC), with chronic exposure being a major risk factor for hepatocellular carcinoma (HCC). HCC is one of the most common and deadly cancers worldwide, particularly in regions with high aflatoxin contamination [31].

Children are particularly vulnerable to the effects of chronic aflatoxin exposure. Studies have shown that aflatoxin ingestion is associated with stunted growth, underweightedness, and developmental delays. These effects are thought to result from a combination of immune suppression, reduced nutrient absorption, and direct toxicity to developing tissues. Addressing aflatoxin contamination is critical for improving child health and development, particularly in low-resource settings [32]. In addition to human health, aflatoxins pose significant risks to livestock and poultry. Chronic exposure to aflatoxin-contaminated feed reduces animal productivity, causing liver damage, reduced growth rates, and increased susceptibility to diseases. High levels of contamination can lead to acute toxicity and mortality, resulting in significant economic losses for farmers [33].

The global burden of aflatoxin-related diseases is substantial, particularly in developing countries where food safety regulations are often inadequate. Aflatoxin exposure contributes to millions of cases of liver cancer and other health conditions annually, highlighting the need for effective prevention and control measures. Public awareness campaigns and education programs are essential for reducing aflatoxin exposure and improving food safety [34].

### 2.4. Other Toxic Metabolites Produced by Aspergillus

In addition to aflatoxins, Aspergillus species produce other toxic metabolites, such as ochratoxin A, sterigmatocystin, and cyclopiazonic acid. Ochratoxin A is nephrotoxic and carcinogenic, with chronic exposure being linked to kidney damage and urinary tract tumors. Sterigmatocystin, a precursor to aflatoxins, exhibits similar toxic properties, including carcinogenicity and mutagenicity [35]. Cyclopiazonic acid is another toxic metabolite produced by Aspergillus species, particularly *Aspergillus flavus*. This compound is neurotoxic and has been implicated in cases of livestock poisoning. The co-occurrence of cyclopiazonic acid with aflatoxins in contaminated food and feed complicates the health risks associated with Aspergillus contamination [36].

The production of these toxic metabolites is influenced by environmental factors such as temperature, humidity, and substrate availability. Optimal conditions for toxin production vary depending on the metabolite, but generally include warm temperatures and high humidity. Understanding these factors is essential for developing targeted mitigation strategies [37]. The toxicity of these metabolites is often synergistic with aflatoxins, amplifying the overall health risks. For example, the combined exposure to aflatoxins and ochratoxin A has been shown to increase the risk of liver and kidney damage. This highlights the need for comprehensive risk assessment and mitigation strategies that address multiple mycotoxins [38].

### 2.5. Synergistic Effects of Multiple Mycotoxins

The co-occurrence of multiple mycotoxins in food and feed can lead to synergistic effects, amplifying their toxicity. For example, the combined exposure to aflatoxins and fumonisins has been shown to increase the risk of liver and esophageal cancer. These interactions complicate risk assessment and mitigation efforts, as the overall toxicity may be greater than the sum of individual effects [39]. Synergistic effects are particularly concerning in regions where multiple mycotoxins are prevalent in staple foods. For example, maize in sub-Saharan Africa is often contaminated with both aflatoxins and fumonisins, posing significant health risks to consumers. Understanding these interactions is essential for developing targeted mitigation strategies [40].

The mechanisms underlying synergistic effects are complex and may involve interactions at the molecular, cellular, and systemic levels. For example, aflatoxins and fumonisins both induce oxidative stress and DNA damage, but through different pathways. Their combined effects may overwhelm cellular defense mechanisms, leading to increased toxicity [41].

## 3. Health Risks Associated with Aflatoxin Exposure

### 3.1. Hepatotoxicity and Liver Cancer

Aflatoxin exposure is a major risk factor for hepatocellular carcinoma (HCC), one of the most common and deadly cancers worldwide (Figure 6). The International Agency for Research on Cancer (IARC) has classified aflatoxin B1 as a Group 1 carcinogen. Chronic exposure to even low levels of aflatoxins can lead to the accumulation of DNA damage, resulting in liver cancer [42]. The primary mechanism of aflatoxin-induced hepatotoxicity involves the formation of DNA adducts, particularly AFB1-N7-guanine. This adduct causes mutations in the p53 tumor suppressor gene, leading to uncontrolled cell proliferation and tumor formation. The liver, being the primary site of aflatoxin metabolism, is particularly vulnerable to these effects [43].

In addition to DNA damage, aflatoxins induce oxidative stress by generating reactive oxygen species (ROS). ROS cause lipid peroxidation, protein oxidation, and DNA damage, leading to cellular dysfunction and apoptosis. The combined effects of DNA adduct formation and oxidative stress exacerbate the hepatotoxic effects of aflatoxins [44]. The risk of aflatoxin-induced liver cancer is influenced by genetic factors, such as polymorphisms in genes involved in aflatoxin metabolism and detoxification (Figure 7). For example, individuals with certain variants of the glutathione S-transferase (GST) gene are more susceptible to aflatoxin-induced liver damage. Understanding these genetic factors is essential for identifying at-risk populations and developing personalized prevention strategies [45].

### 3.2. Immunosuppression and Increased Susceptibility to Infections

Aflatoxins impair both innate and adaptive immunity, reducing the body’s ability to fight infections. This immunosuppressive effect is particularly concerning in regions with a high prevalence of infectious diseases, such as HIV/AIDS and malaria. Studies have shown that aflatoxin exposure is associated with increased viral load and disease progression in HIV-positive individuals [46]. The mechanisms underlying aflatoxin-induced immunosuppression involve the suppression of cytokine production and antibody responses. Aflatoxins inhibit the proliferation of immune cells, such as T cells and B cells, reducing their ability to mount an effective immune response. This increases susceptibility to infections and complicates disease management [47]. Children are particularly vulnerable to the immunosuppressive effects of aflatoxins. Chronic exposure to aflatoxin-contaminated food is associated with increased incidence of infectious diseases, such as diarrhea and respiratory infections. These effects are thought to result from a combination of immune suppression and reduced nutrient absorption [48].

### 3.3. Growth Impairment and Developmental Delays in Children

Children are particularly vulnerable to the effects of aflatoxin exposure, which has been linked to stunted growth, underweightedness, and developmental delays. These effects are thought to result from a combination of immune suppression, reduced nutrient absorption, and direct toxicity to developing tissues. Addressing aflatoxin contamination is critical for improving child health and development [49]. The mechanisms underlying aflatoxin-induced growth impairment involve the disruption of nutrient metabolism and absorption. Aflatoxins impair the function of the gastrointestinal tract, reducing the absorption of essential nutrients such as proteins, vitamins, and minerals. This leads to malnutrition and stunted growth, particularly in children with chronic exposure [50]. In addition to physical growth, aflatoxin exposure is associated with cognitive and developmental delays. Studies have shown that children exposed to high levels of aflatoxins perform poorly on cognitive tests and have lower school attendance rates. These effects are thought to result from the neurotoxic properties of aflatoxins and their impact on brain development [51].

### 3.4. Economic Impact on Livestock and Poultry

Aflatoxin-contaminated feed reduces the productivity and profitability of livestock and poultry farming. In animals, aflatoxins cause liver damage, reduced growth rates, and increased susceptibility to diseases. High levels of contamination can lead to acute toxicity and mortality, resulting in significant economic losses for farmers [52]. The mechanisms underlying aflatoxin-induced toxicity in animals are similar to those in humans, involving DNA damage, oxidative stress, and immune suppression. Aflatoxins impair the function of the liver, reducing its ability to metabolize nutrients and detoxify harmful substances. This leads to reduced growth rates and increased susceptibility to infections [53]. In addition to direct health effects, aflatoxin contamination affects the quality of animal products, such as milk, meat, and eggs. For example, aflatoxin M1, a metabolite of aflatoxin B1, is excreted in milk and poses risks to human consumers. This highlights the need for effective mitigation strategies to protect both animal and human health [54].

### 3.5. Global Burden of Aflatoxin-Related Diseases

The global burden of aflatoxin-related diseases is substantial, particularly in developing countries where food safety regulations are often inadequate (Table 3). Aflatoxin exposure contributes to millions of cases of liver cancer and other health conditions annually, highlighting the need for effective prevention and control measures [55].

The economic impact of aflatoxin-related diseases is also significant, with billions of dollars lost annually due to healthcare costs, reduced productivity, and trade restrictions. In developing countries, where resources are limited, the burden of aflatoxin-related diseases exacerbates poverty and food insecurity [56].

## 4. Mitigation Strategies for Aflatoxin Contamination

### 4.1. Pre-Harvest Interventions

Pre-harvest strategies are critical for reducing fungal growth and aflatoxin production in the field. One of the most effective approaches is the use of resistant crop varieties, which are less susceptible to Aspergillus infection (Table 4). Breeding programs have developed maize and peanut varieties with enhanced resistance to aflatoxin contamination, offering a sustainable solution for farmers in high-risk regions [57]. For instance, the maize hybrid ‘GAF4′ developed in Ghana has demonstrated reduced aflatoxin accumulation under field conditions. Similarly, in the United States, conventional breeding has led to peanut cultivars such as ‘Georganic’ and ‘Tifguard’ that show improved resistance to aflatoxin contamination. Another key pre-harvest intervention is the application of biocontrol agents, such as non-toxigenic strains of *Aspergillus flavus*. A classical example is the use of the non-aflatoxigenic strain AF36 in the U.S. cotton industry, which has significantly reduced aflatoxin levels in cottonseed. More recently, in Nigeria, the biocontrol product Aflasafe^®^ has been successfully applied on maize and groundnuts, leading to aflatoxin reductions of up to 80%. These strains competitively inhibit the growth of toxigenic strains, reducing aflatoxin production. Field trials in Africa have demonstrated the effectiveness of this approach, with significant reductions in aflatoxin levels observed in treated crops [58].

Table 4 Description of Mitigation Strategies for Aflatoxin Contamination.

Good agricultural practices (GAPs) also play a crucial role in minimizing aflatoxin contamination. Practices such as crop rotation, timely harvesting, and proper irrigation can reduce fungal growth and aflatoxin production. For example, rotating maize with legumes can disrupt the life cycle of Aspergillus species, reducing their prevalence in the soil [59]. Climate-smart strategies are increasingly being explored to mitigate the impact of climate change on aflatoxin contamination. Drought-resistant crop varieties and improved water management practices can help reduce stress on crops, making them less susceptible to fungal infection. These strategies are particularly important in regions where climate change is expected to exacerbate aflatoxin contamination [60].

Integrated pest management (IPM) is another effective pre-harvest strategy for reducing aflatoxin contamination. IPM combines biological, cultural, and chemical methods to control pests and diseases, reducing the risk of fungal infection. For example, the use of insect-resistant crops can reduce insect damage, which often facilitates Aspergillus infection [61]. Public awareness and farmer education programs are essential for the successful implementation of pre-harvest interventions. Farmers need to be informed about the risks of aflatoxin contamination and the importance of adopting good agricultural practices. Extension services and training programs can play a key role in disseminating this knowledge [62].

### 4.2. Post-Harvest Management

Post-harvest measures are critical for preventing aflatoxin contamination during storage and processing. The proper drying of crops is one of the most effective ways to reduce fungal growth. Moisture levels should be maintained below 13% to inhibit Aspergillus growth and aflatoxin production. Traditional sun drying on raised platforms has long been used in parts of India and Southeast Asia, while recent innovations such as solar bubble dryers in Bangladesh and hermetic drying bags in Kenya have improved moisture control during storage. Mechanical dryers and solar drying techniques are commonly used to achieve this [63]. Storage conditions also play a crucial role in preventing aflatoxin contamination. Crops should be stored in clean, dry, and well-ventilated facilities to minimize fungal growth. The use of Purdue Improved Crop Storage (PICS) bags in West Africa is a notable example. These triple-layer hermetic bags have shown effectiveness in limiting fungal proliferation and aflatoxin production in stored grains such as maize and cowpea. Hermetic storage systems, which create an oxygen-free environment, have been shown to effectively reduce aflatoxin levels in stored crops [64].

Chemical treatments, such as the application of ozone and ammonia, are effective in degrading aflatoxins in contaminated crops. Ozone treatment has been shown to reduce aflatoxin levels by up to 90% in maize and peanuts, while ammonia treatment can detoxify aflatoxins in animal feed. However, these methods must be carefully regulated to ensure food safety [65]. A classical example includes the ammoniation of contaminated maize in the United States, which was implemented under FDA guidance for feed detoxification. Recently, in China, ozonation has been scaled in peanut processing facilities, with documented reductions in aflatoxin B1 without compromising nutritional quality. Physical methods, such as sorting and cleaning, are also effective in reducing aflatoxin levels in contaminated crops. Manual or mechanical sorting can remove visibly moldy or damaged kernels, which are more likely to contain high levels of aflatoxins. A well-documented classical example is the hand-sorting of groundnuts in Senegal, which has historically reduced aflatoxin content by up to 70%. In recent years, machine vision-based optical sorting technologies have been deployed in European nut and grain industries for automated mycotoxin risk reduction. This approach is particularly useful in resource-limited settings [66].

Irradiation and UV treatment are emerging technologies for reducing aflatoxin contamination in food and feed. Gamma irradiation has been shown to degrade aflatoxins in maize and peanuts, while UV treatment can reduce aflatoxin levels in milk and dairy products. These methods offer a non-chemical approach to aflatoxin mitigation [67]. Public awareness and education programs are essential for the successful implementation of post-harvest interventions. Farmers and food processors need to be informed about the importance of proper drying, storage, and handling practices to minimize aflatoxin contamination. Extension services and training programs can play a key role in disseminating this knowledge [68].

### 4.3. Biological Control Methods

Biological control methods offer sustainable solutions for aflatoxin management. The use of non-toxigenic strains of *Aspergillus flavus* is one of the most effective biological control strategies. These strains competitively inhibit the growth of toxigenic strains, reducing aflatoxin production. Field trials in Africa and the United States have demonstrated the effectiveness of this approach [69]. Enzymatic degradation of aflatoxins is another promising biological control method. Certain micro-organisms produce enzymes that can degrade aflatoxins into non-toxic compounds. For example, *Flavobacterium aurantiacum* produces an enzyme that degrades AFB1 into less toxic metabolites. This approach offers a potential solution for detoxifying aflatoxin-contaminated food and feed [70].

The use of microbial antagonists, such as bacteria and fungi, is another effective biological control strategy. These micro-organisms can inhibit the growth of Aspergillus species and reduce aflatoxin production. For example, *Bacillus subtilis* and *Trichoderma harzianum* have been shown to reduce aflatoxin levels in maize and peanuts [71]. Plant-based compounds, such as essential oils and plant extracts, have also been explored for their potential to inhibit Aspergillus growth and aflatoxin production. For example, neem oil and clove oil have been shown to reduce aflatoxin levels in stored crops. These natural compounds offer a safe and environmentally friendly alternative to chemical treatments [72].

Integrated biological control approaches, which combine multiple strategies, are often more effective than single methods. For example, combining non-toxigenic strains of *Aspergillus flavus* with microbial antagonists can provide enhanced protection against aflatoxin contamination. This approach offers a sustainable solution for reducing aflatoxin levels in food and feed [73]. Public awareness and education programs are essential for the successful implementation of biological control methods. Farmers need to be informed about the benefits of biological control and the importance of adopting these practices. Extension services and training programs can play a key role in disseminating this knowledge [74].

### 4.4. Physical and Chemical Detoxification

Physical methods, such as sorting and cleaning, are effective in reducing aflatoxin levels in contaminated crops. Manual or mechanical sorting can remove visibly moldy or damaged kernels, which are more likely to contain high levels of aflatoxins. This approach is particularly useful in resource-limited settings [75]. Irradiation and UV treatment are emerging technologies for reducing aflatoxin contamination in food and feed. Gamma irradiation has been shown to degrade aflatoxins in maize and peanuts, while UV treatment can reduce aflatoxin levels in milk and dairy products. These methods offer a non-chemical approach to aflatoxin mitigation [76].

Chemical detoxification methods, such as the application of ozone and ammonia, are effective in degrading aflatoxins in contaminated crops. Ozone treatment has been shown to reduce aflatoxin levels by up to 90% in maize and peanuts, while ammonia treatment can detoxify aflatoxins in animal feed. However, these methods must be carefully regulated to ensure food safety [77]. The use of adsorbents, such as activated charcoal and clay minerals, is another effective chemical detoxification method. These materials can bind aflatoxins in the gastrointestinal tract, reducing their absorption and toxicity. For example, bentonite clay has been shown to reduce aflatoxin levels in animal feed [78].

Integrated detoxification approaches, which combine physical and chemical methods, are often more effective than single methods. For example, combining sorting with ozone treatment can provide enhanced protection against aflatoxin contamination. This approach offers a comprehensive solution for reducing aflatoxin levels in food and feed [79]. Public awareness and education programs are essential for the successful implementation of physical and chemical detoxification methods. Farmers and food processors need to be informed about the benefits of these methods and the importance of adopting them. Extension services and training programs can play a key role in disseminating this knowledge [80].

### 4.5. Integrated Management Approaches

An integrated approach combining pre-harvest, post-harvest, and biological control methods is essential for effective aflatoxin management. For example, combining resistant crop varieties with biocontrol agents and proper storage practices can provide comprehensive protection against aflatoxin contamination. This approach has been successfully implemented in several countries, including Kenya and the United States [81]. Integrated pest management (IPM) is another effective strategy for reducing aflatoxin contamination. IPM combines biological, cultural, and chemical methods to control pests and diseases, reducing the risk of fungal infection. For example, the use of insect-resistant crops can reduce insect damage, which often facilitates Aspergillus infection [82].

Climate-smart strategies, such as drought-resistant crops and improved water management practices, are increasingly being integrated into aflatoxin management programs. These strategies help reduce stress on crops, making them less susceptible to fungal infection. This is particularly important in regions where climate change is expected to exacerbate aflatoxin contamination [83]. Public awareness and education programs are essential for the successful implementation of integrated management approaches. Farmers need to be informed about the benefits of these approaches and the importance of adopting them. Extension services and training programs can play a key role in disseminating this knowledge [84].

## 5. Chemical Analysis of Aflatoxins and Toxic Metabolites

### 5.1. Sample Preparation Techniques

Accurate detection of aflatoxins requires efficient sample preparation techniques to isolate these toxins from complex food matrices. Solvent extraction is one of the most common methods, where organic solvents like methanol or acetonitrile are used to extract aflatoxins from samples. This method is effective and may require additional cleanup steps to remove interfering compounds [85]. Solid-phase extraction (SPE) is another widely used technique for sample preparation. SPE involves passing the sample through a cartridge packed with adsorbent material, which selectively binds aflatoxins. This method offers high recovery rates and is particularly useful for analyzing low-concentration samples. However, it requires careful optimization to ensure reproducibility [86].

Immunoaffinity column cleanup is a highly specific sample preparation method that uses antibodies to selectively bind aflatoxins. This technique is particularly effective for removing matrix interferences, making it suitable for complex food samples like maize and peanuts. Immunoaffinity columns are often used in conjunction with chromatographic methods for accurate quantification [87]. Emerging sample preparation techniques, such as QuEChERS (Quick, Easy, Cheap, Effective, Rugged, and Safe), are gaining popularity due to their simplicity and efficiency. QuEChERS involves a two-step process of extraction and cleanup, using a combination of salts and adsorbents. This method is particularly useful for multi-mycotoxin analysis, as it can simultaneously extract multiple toxins from a single sample [88].

### 5.2. Chromatographic Methods

High-performance liquid chromatography (HPLC) is one of the most widely used chromatographic methods for aflatoxin analysis. HPLC separates aflatoxins based on their chemical properties and detects them using ultraviolet (UV) or fluorescence detectors. This method offers high sensitivity and specificity, making it suitable for quantifying aflatoxins in complex food matrices [89]. Liquid chromatography-tandem mass spectrometry (LC-MS/MS) is another powerful technique for aflatoxin analysis. LC-MS/MS combines the separation capabilities of liquid chromatography with the detection power of mass spectrometry, allowing for the simultaneous analysis of multiple aflatoxins and their metabolites. This method is particularly useful for detecting low-concentration aflatoxins in complex samples [90].

Gas chromatography-mass spectrometry (GC-MS) is less commonly used for aflatoxin analysis due to the need for derivatization, which makes the process more complex. However, GC-MS is highly sensitive and can be used for specific applications, such as analyzing volatile aflatoxin metabolites. This method is often used in research settings where high sensitivity is required [91]. Ultra-performance liquid chromatography (UPLC) is an advanced version of HPLC that offers faster analysis times and higher resolution. UPLC is particularly useful for high-throughput laboratories, where rapid analysis of large numbers of samples is required. This method has been successfully applied to the analysis of aflatoxins in various food matrices [92].

### 5.3. Immunoassays and Rapid Screening Methods

Enzyme-linked immunosorbent assays (ELISA) are widely used for the rapid screening of aflatoxins. ELISA kits are commercially available and offer a cost-effective and user-friendly solution for detecting aflatoxins in food and feed. These kits use antibodies to specifically bind aflatoxins, allowing for their detection using colorimetric or fluorescent signals [93]. Lateral flow devices (LFDs) are another rapid screening method that provides on-site detection of aflatoxins. LFDs are portable, easy to use, and provide results within minutes. These devices are particularly useful for field testing, where quick decisions about food safety are required. However, LFDs are generally less sensitive than laboratory-based methods [94].

Immunochromatographic assays (ICAs) are similar to LFDs but offer higher sensitivity and specificity. ICAs use gold nanoparticles or other labels to enhance the detection signal, making them suitable for detecting low-concentration aflatoxins. These assays are often used for screening large numbers of samples in resource-limited settings [95]. Fluorescence polarization immunoassays (FPIAs) are another rapid screening method that uses fluorescently labeled antibodies to detect aflatoxins. FPIAs offer high sensitivity and can be performed in a matter of minutes. This method is particularly useful for high-throughput screening of food and feed samples [96].

### 5.4. Emerging Technologies for Aflatoxin Detection

Biosensors are emerging as a promising technology for aflatoxin detection. These devices use biological recognition elements, such as antibodies or enzymes, to specifically bind aflatoxins and generate a measurable signal. Biosensors offer high sensitivity, portability, and real-time detection capabilities, making them suitable for on-site testing [97]. Nanotechnology-based platforms are also being explored for aflatoxin detection. For example, gold nanoparticles and quantum dots can be used to enhance the sensitivity of detection methods. These nanomaterials offer unique optical and electronic properties that can be leveraged for the development of highly sensitive and selective detection systems [98].

Spectroscopic methods, such as near-infrared (NIR) spectroscopy and Raman spectroscopy, are non-destructive techniques that can be used for the rapid detection of aflatoxins. These methods analyze the interaction of light with the sample to identify specific chemical signatures associated with aflatoxins. Spectroscopic methods are particularly useful for the high-throughput screening of food and feed samples [99]. Molecularly imprinted polymers (MIPs) are synthetic materials that can selectively bind aflatoxins. MIPs are often used in conjunction with other detection methods, such as chromatography or spectroscopy, to enhance sensitivity and selectivity. These materials offer a cost-effective and stable alternative to biological recognition elements [100].

### 5.5. Challenges in Aflatoxin Analysis

One of the major challenges in aflatoxin analysis is the presence of matrix effects, which can interfere with the detection and quantification of aflatoxins. Matrix effects are particularly problematic in complex food matrices, such as maize and peanuts, where co-extracted compounds can affect the accuracy of the analysis. Advanced sample preparation techniques and detection methods are required to minimize these effects [101]. The co-occurrence of multiple mycotoxins in food and feed samples is another challenge in aflatoxin analysis. Many analytical methods are designed to detect a single mycotoxin, making it difficult to accurately quantify multiple toxins in a single sample. Multi-mycotoxin methods, such as LC-MS/MS, are increasingly being used to address this challenge [102]. The need for method validation and standardization is another critical issue in aflatoxin analysis. Different laboratories may use different methods and protocols, leading to variability in results. Standardized methods and reference materials are essential for ensuring the accuracy and reproducibility of aflatoxin analysis [103].

## 6. Regulatory Framework and Global Perspectives

### 6.1. International Regulations and Standards

International regulatory agencies, such as the World Health Organization (WHO) and the Food and Agriculture Organization (FAO), have established permissible limits for aflatoxins in food and feed. The Codex Alimentarius Commission sets global standards, which serve as a reference for national regulations. For example, the maximum permissible limit for aflatoxin B1 in food is set at 2 µg/kg in the European Union, while the limit in the United States is 20 µg/kg for animal feed [104]. Developing countries often face challenges in aligning their national regulations with international standards due to limited resources and infrastructure. This discrepancy can lead to trade disputes and economic losses, as crops exceeding permissible limits are rejected by importing countries. Strengthening regulatory frameworks and capacity building are essential for improving compliance and ensuring food safety [105]. Harmonization of international regulations is critical for reducing the global burden of aflatoxin contamination. Collaborative efforts between countries and international organizations can help establish uniform standards and facilitate trade. Public awareness campaigns and education programs are also essential for promoting compliance with regulatory standards [106].

### 6.2. Challenges in Developing Countries

Developing countries face significant challenges in implementing and enforcing aflatoxin regulations due to limited resources and inadequate infrastructure. For example, many countries lack the laboratory facilities and trained personnel needed for accurate aflatoxin analysis. This makes it difficult to monitor and control aflatoxin contamination in food and feed [107]. Another major challenge is the lack of awareness among farmers and food processors about the risks of aflatoxin contamination. Many small-scale farmers in developing countries are unaware of good agricultural practices and post-harvest management techniques that can reduce aflatoxin levels. Public awareness campaigns and training programs are essential for addressing this issue [108]. Strengthening regulatory frameworks and capacity building are critical for improving compliance with aflatoxin regulations in developing countries. International organizations and NGOs play a key role in providing technical assistance and financial support to help countries develop and implement effective regulatory systems [109].

### 6.3. Role of International Organizations

International organizations, such as the WHO, FAO, and the International Agency for Research on Cancer (IARC), play a critical role in addressing aflatoxin contamination. These organizations conduct research, develop guidelines, and provide technical assistance to help countries implement effective aflatoxin control measures. For example, the FAO has developed a comprehensive framework for aflatoxin management, which includes pre-harvest, post-harvest, and regulatory interventions [110]. Collaborative efforts between international organizations, governments, and the private sector are essential for achieving global food safety goals. For example, the Partnership for Aflatoxin Control in Africa (PACA) is a multi-stakeholder initiative that aims to reduce aflatoxin contamination in Africa through research, policy development, and capacity building. Such initiatives provide a platform for sharing knowledge and resources to address the global challenge of aflatoxin contamination [111]. International organizations also play a key role in raising awareness about the risks of aflatoxin contamination and promoting compliance with regulatory standards. Public awareness campaigns and education programs are essential for reducing aflatoxin exposure and improving food safety. These efforts are particularly important in developing countries, where the burden of aflatoxin-related diseases is highest [112].

### 6.4. Trade Implications and Economic Impact

Aflatoxin contamination has significant implications for international trade, as many countries have strict regulations on permissible aflatoxin levels in imported food and feed. Crops exceeding these limits are often rejected, leading to significant economic losses for exporting countries. For example, African countries lose an estimated $670 million annually due to aflatoxin-related trade restrictions [113]. The economic impact of aflatoxin contamination extends beyond trade losses. Contaminated crops reduce the productivity and profitability of agriculture, particularly in developing countries where small-scale farmers are most affected. Aflatoxin contamination also increases healthcare costs due to the high prevalence of aflatoxin-related diseases, such as liver cancer and immune suppression [114]. Efforts to reduce the economic impact of aflatoxin contamination must address both trade and public health issues. Strengthening regulatory frameworks, improving agricultural practices, and promoting international collaboration are essential for reducing aflatoxin levels and ensuring food safety. Public awareness campaigns and education programs are also critical for reducing aflatoxin exposure and improving public health [115].

### 6.5. Public Awareness and Education

Public awareness campaigns and education programs are essential for reducing aflatoxin exposure and improving food safety. Farmers, consumers, and policymakers must be informed about the risks of aflatoxin contamination and the importance of adopting good agricultural practices and post-harvest management techniques [116]. Extension services and training programs play a key role in disseminating knowledge about aflatoxin control measures. For example, the FAO has developed training modules for farmers and food processors on aflatoxin management. These programs provide practical guidance on reducing aflatoxin levels in crops and ensuring food safety [117]. Public awareness campaigns are also essential for promoting compliance with regulatory standards and reducing aflatoxin exposure. These campaigns can be conducted through various channels, including radio, television, and social media. By raising awareness about the risks of aflatoxin contamination, these campaigns help improve food safety and public health [118].

## 7. Future Perspectives and Research Directions

### 7.1. Genetic Engineering and Crop Improvement

Genetic engineering offers promising solutions for developing aflatoxin-resistant crop varieties. Advances in CRISPR-Cas9 technology and genome editing are being explored to enhance crop resistance to Aspergillus infection and aflatoxin production. For example, researchers have successfully engineered maize varieties with reduced susceptibility to aflatoxin contamination [119]. The development of aflatoxin-resistant crops requires a deep understanding of the genetic and biochemical mechanisms underlying Aspergillus infection and aflatoxin production. Omics technologies, such as genomics and proteomics, provide valuable insights into these mechanisms, enabling the identification of key genes and pathways involved in aflatoxin resistance [120].

In addition to genetic engineering, traditional breeding methods can also be used to develop aflatoxin-resistant crop varieties. For example, marker-assisted selection (MAS) allows breeders to identify and select plants with desirable traits, such as resistance to Aspergillus infection. Combining genetic engineering with traditional breeding methods offers a comprehensive approach to crop improvement [121]. Efforts to develop aflatoxin-resistant crops must also consider the potential impact on crop yield and quality. For example, some genetic modifications may reduce aflatoxin levels but also affect the nutritional value or taste of the crop. Balancing aflatoxin resistance with other desirable traits is essential for ensuring the success of crop improvement programs [122].

### 7.2. Nanotechnology in Aflatoxin Management

Nanotechnology-based approaches are emerging as innovative solutions for aflatoxin management. For example, nano-encapsulation of biocontrol agents, such as non-toxigenic strains of *Aspergillus flavus*, can enhance their stability and effectiveness in reducing aflatoxin production. These nano-formulations offer a sustainable and environmentally friendly alternative to chemical treatments [123]. Nano-sensors are another promising application of nanotechnology in aflatoxin management. These devices use nanomaterials, such as gold nanoparticles and quantum dots, to detect aflatoxins with high sensitivity and specificity. Nano-sensors offer a portable and cost-effective solution for on-site aflatoxin detection, making them suitable for use in resource-limited settings [124]. Nanotechnology can also be used to develop novel detoxification methods for aflatoxin-contaminated food and feed. For example, nanoparticles can be engineered to bind aflatoxins and remove them from contaminated products. These nano-based detoxification methods offer a safe and effective alternative to traditional chemical treatments [125].

### 7.3. Integration of Omics Technologies

Omics technologies, including genomics, proteomics, and metabolomics, provide valuable insights into the biology of Aspergillus and the mechanisms of aflatoxin production. For example, genomics can be used to identify key genes involved in aflatoxin biosynthesis, while proteomics can reveal the protein interactions that regulate this process [126]. Metabolomics is another powerful tool for studying aflatoxin production. This technology analyzes the metabolic pathways and compounds involved in aflatoxin biosynthesis, providing a comprehensive understanding of the biochemical processes underlying aflatoxin contamination. Metabolomics can also be used to identify biomarkers for early detection of aflatoxin contamination [127]. Integrating omics technologies with other approaches, such as genetic engineering and biocontrol, can lead to the development of targeted mitigation strategies. For example, genomics and proteomics can be used to identify potential targets for genetic modification or biocontrol agents, while metabolomics can provide insights into the effectiveness of these interventions [128].

### 7.4. Climate Change and Aflatoxin Contamination

Climate change is expected to exacerbate aflatoxin contamination by creating favorable conditions for fungal growth. Rising temperatures and changing precipitation patterns can increase the prevalence of Aspergillus species and aflatoxin production in crops. This poses a significant threat to food security, particularly in regions where aflatoxin contamination is already a major issue [129]. Efforts to mitigate the impact of climate change on aflatoxin contamination must focus on developing climate-resilient crops and adaptive agricultural practices. For example, drought-resistant crop varieties and improved water management techniques can help reduce stress on crops, making them less susceptible to fungal infection [130]. Research is also needed to understand the complex interactions between climate change, fungal growth, and aflatoxin production. For example, studies have shown that elevated CO2 levels can increase aflatoxin production in some crops. Understanding these interactions is essential for developing effective mitigation strategies [131].

### 7.5. Global Collaboration and Capacity Building

Addressing the global challenge of aflatoxin contamination requires collaboration between researchers, policymakers, and stakeholders. International organizations, such as the WHO and FAO, play a key role in facilitating this collaboration by providing technical assistance, funding, and platforms for knowledge sharing [132]. Capacity building is essential for improving aflatoxin management in developing countries. This includes training programs for farmers and food processors, as well as investments in laboratory infrastructure and regulatory systems. International organizations and NGOs play a critical role in providing the resources and expertise needed for capacity building [133]. Public-private partnerships are another important avenue for addressing aflatoxin contamination. For example, collaborations between research institutions, governments, and the private sector can lead to the development of innovative solutions, such as aflatoxin-resistant crops and advanced detection methods. These partnerships provide a platform for sharing knowledge and resources to address the global challenge of aflatoxin contamination [134].

## 8. Conclusions

Aflatoxins and other toxic metabolites produced by *Aspergillus* species pose a major threat to global food safety, public health, and economic stability. Their widespread contamination of staple crops, particularly in warm and humid regions, contributes to severe health consequences such as hepatocellular carcinoma, immune suppression, and growth impairment in children. Additionally, the economic impact of aflatoxin contamination, including trade restrictions and losses in agricultural productivity, further exacerbates food insecurity in developing countries. Addressing aflatoxin contamination requires a multifaceted approach that integrates pre-harvest and post-harvest management strategies, biological control methods, and chemical detoxification techniques. Advances in analytical techniques, including chromatographic, immunoassay, and biosensor-based methods, play a critical role in ensuring food safety and regulatory compliance. However, challenges such as climate change, weak regulatory enforcement in developing regions, and the co-occurrence of multiple mycotoxins necessitate continued research and global collaboration. Future efforts should focus on the development of aflatoxin-resistant crop varieties through genetic engineering, the application of nanotechnology for detection and detoxification, and the adoption of climate-smart agricultural practices. Strengthening regulatory frameworks, enhancing surveillance systems, and increasing public awareness are also essential for reducing aflatoxin exposure and its associated risks. By combining scientific innovation, policy interventions, and international cooperation, sustainable solutions can be achieved to mitigate the impact of aflatoxins on food security and public health worldwide.

## Figures and Tables

**Figure 1 toxins-17-00331-f001:**
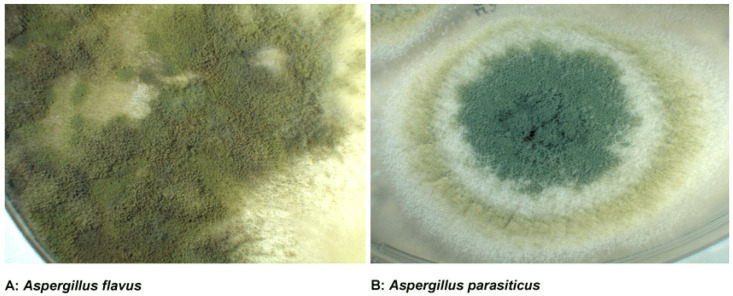
Aspergillus species.

**Figure 2 toxins-17-00331-f002:**
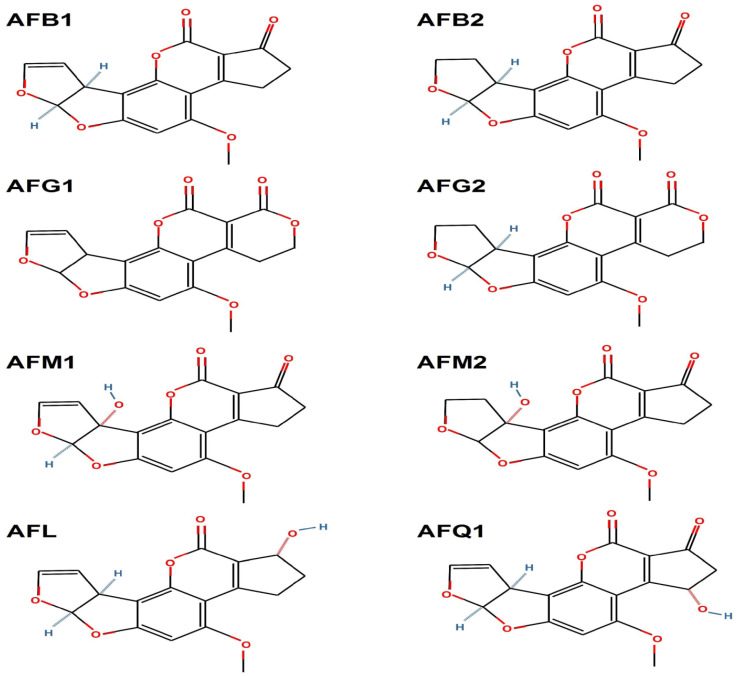
Chemical structure of different types of aflatoxins.

**Figure 3 toxins-17-00331-f003:**
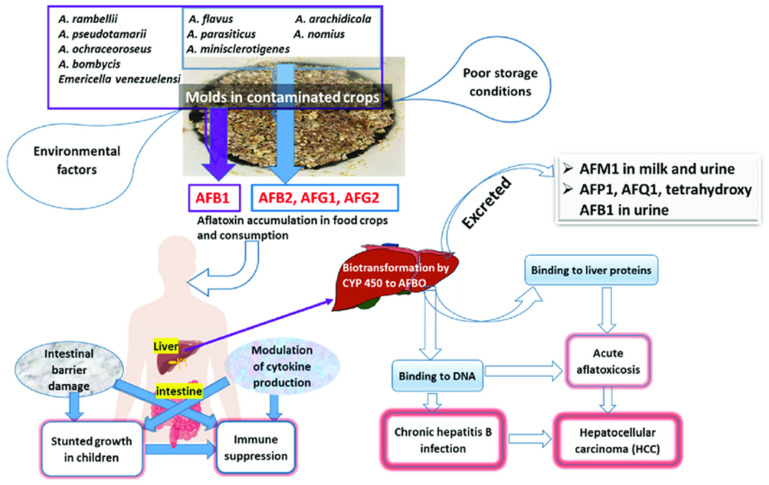
Factors influencing aflatoxin toxicity.

**Figure 4 toxins-17-00331-f004:**
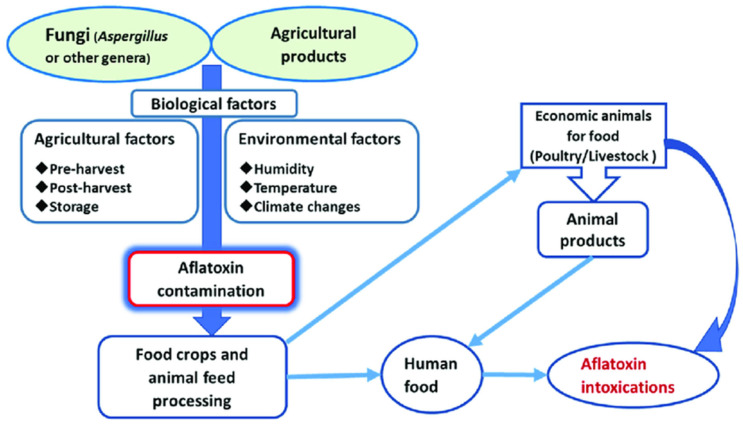
Formation of aflatoxin intoxifications.

**Figure 5 toxins-17-00331-f005:**
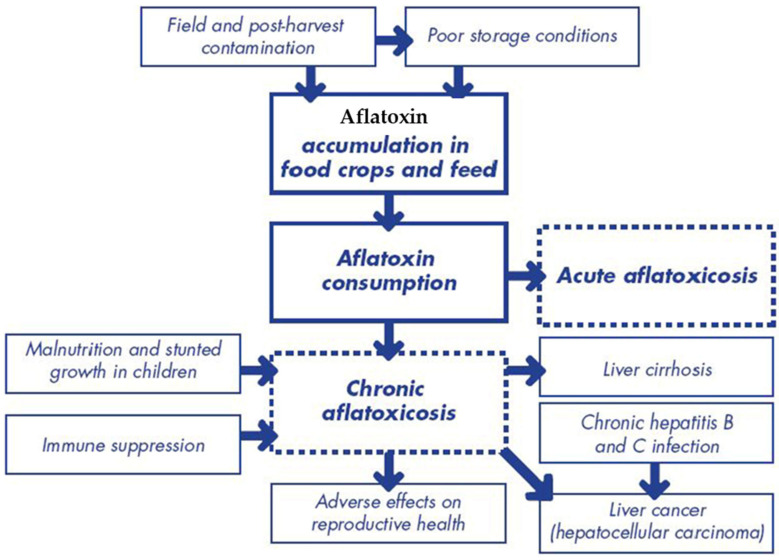
Acute aflatoxin exposure to cause aflatoxicosis.

**Figure 6 toxins-17-00331-f006:**
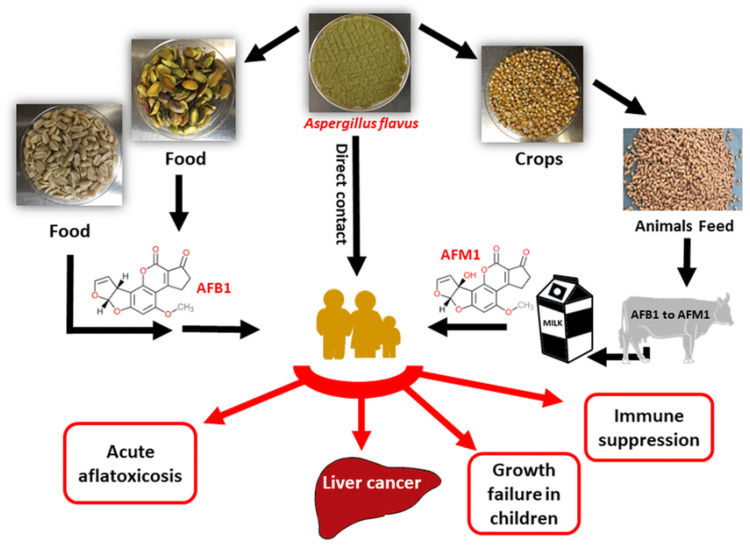
Health Risks Associated with Aflatoxin Exposure.

**Figure 7 toxins-17-00331-f007:**
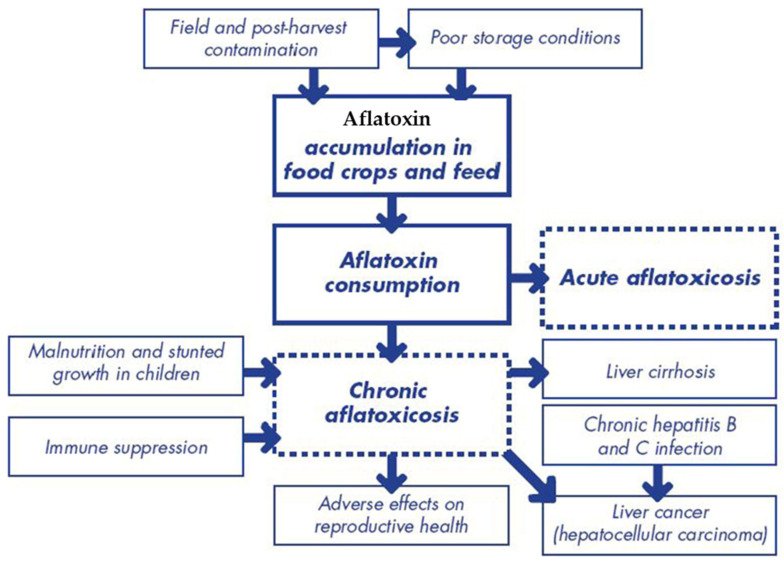
Hepatotoxicity and liver cancer by aflatoxin exposure.

**Table 1 toxins-17-00331-t001:** Economic and Public Health Impact of Aflatoxin Contamination [9].

Impact Area	Public Health Impact	Economic Impact
Health Risks	- Carcinogenic (liver cancer).	- Reduced crop marketability due to contamination.
	- Acute aflatoxicosis (vomiting, abdominal pain, and death, in severe cases).	- Loss of income for farmers due to crop rejection and loss.
Chronic Exposure	- Immune suppression.	- Increased healthcare costs for treating liver diseases and cancers.
	- Growth impairment (especially in children).	- Loss of export opportunities due to international trade restrictions.
Childhood Stunting	- Linked to developmental delay and cognitive impairment.	- Long-term productivity losses due to stunted growth.
Crop Losses	- Aflatoxin contamination reduces crop quality and safety.	- Reduced agricultural yields due to affected crops.
Trade Restrictions	- Risk of public health crises in contaminated regions	- Rejection of aflatoxin-contaminated crops at international borders.
Healthcare Burden	- Increased cases of liver diseases and cancers.	- Increased strain on public health systems.
Case Study Examples	- In Nigeria, interventions reduce liver cancer cases.	- Pre-harvest biocontrol measures in Nigeria show cost-effective results.
Mitigation Strategies	- Promoting awareness about aflatoxins and safe food handling.	- Supporting improved crop storage, pest-resistant crops, and proper harvesting.
Agricultural Impact	- Health impacts, especially in rural communities.	- Financial burden on farmers and the agricultural sector.

**Table 2 toxins-17-00331-t002:** Global Distribution and Prevalence of Aflatoxin Contamination [17].

Region	Countries Affected	Common Contaminated Crops	Prevalence	Key Factors
Sub-Saharan Africa	Kenya, Nigeria, Ghana, Tanzania, Malawi, and Benin.	Maize, peanuts, and sorghum.	High prevalence; >50% of maize samples exceed regulatory limits in some areas.	Warm, humid climates; poor storage practices; limited regulatory enforcement.
Southeast Asia	India, Indonesia, Philippines, Vietnam, and Thailand.	Maize, rice, and peanuts.	High prevalence; frequent contamination in maize and peanuts.	Tropical climate; high humidity; inadequate post-harvest management.
Latin America	Argentina, Brazil, Mexico, and Honduras.	Maize, peanuts, and tree nuts.	Moderate to high prevalence; significant contamination in maize and peanuts.	Warm climates; variable agricultural practices; some regulatory challenges.
North America	United States (Southern states).	Maize, peanuts, and tree nuts.	Low to moderate prevalence; localized outbreaks in drought-affected areas.	Advanced agricultural practices; strict regulatory standards.
Europe	Serbia, Italy, Spain, and Romania.	Maize and cereals.	Low prevalence; occasional contamination due to climate change.	Strict regulatory enforcement; advanced storage and monitoring systems.
Middle East	Iran, Egypt, and Turkey.	Maize, pistachios, and figs.	Moderate prevalence; contamination in pistachios and figs.	Warm climates; variable storage conditions; limited regulatory enforcement.
East Asia	China and South Korea.	Maize, rice, and peanuts.	Moderate prevalence; contamination in maize and peanuts.	Variable climate; improving regulatory standards.

**Table 3 toxins-17-00331-t003:** Global Burden of Aflatoxin-Related Diseases [55].

Disease	Associated Health Impact	Global Incidence (Estimate)	Regions Most Affected	Primary Risk Factors
Liver Cancer (Hepatocellular Carcinoma—HCC)	Aflatoxin exposure is a leading cause of liver cancer.	250,000–500,000 new cases annually.	Sub-Saharan Africa, Southeast Asia, and China.	Chronic exposure to aflatoxin, particularly in combination with HBV infection.
Acute Aflatoxicosis	Acute poisoning leading to liver damage, and in severe cases, death.	10,000–20,000 cases annually.	Asia, Africa, and Latin America.	Consumption of aflatoxin-contaminated food (high levels).
Stunted Growth in Children	Aflatoxin exposure impairs childhood development and growth.	Millions of children at risk.	Sub-Saharan Africa, Southeast Asia.	Chronic exposure to aflatoxins in food over extended periods.
Immune Suppression	Impaired immune system, leading to increased vulnerability to infections.	Significant regional impact.	Africa, Southeast Asia.	Long-term exposure to low levels of aflatoxins.
Liver Cirrhosis	Chronic liver damage and fibrosis due to prolonged exposure.	Thousands of cases annually.	High exposure areas such as sub-Saharan Africa.	Long-term exposure to aflatoxins.
Increased Mortality in HIV Patients	Aflatoxin exacerbates liver disease progression in HIV-infected individuals.	Unclear, but significant.	Sub-Saharan Africa.	Co-exposure to aflatoxins and HIV infection.

**Table 4 toxins-17-00331-t004:** Mitigation Strategies for Aflatoxin Contamination [57].

Pre-Harvest Interventions	- Resistant Cultivars: Use of plant varieties less susceptible to aflatoxin-producing fungi. - Agronomic Practices: Techniques such as optimal planting dates, crop rotation, and irrigation to reduce fungal contamination. - Biological Control: Applying non-aflatoxigenic strains of *Aspergillus* to suppress aflatoxin-producing strains in the field.
Post-Harvest Management	- Sorting and Cleaning: Removal of contaminated grains and debris to lower aflatoxin levels. - Proper Drying and Storage: Ensuring crops are dried properly and stored in conditions that limit fungal growth and mycotoxin production.
Biological Control Methods	- Microbial Application: Introducing non-toxigenic strains of *Aspergillus* to outcompete and suppress aflatoxin-producing strains in the field. - Microbial Detoxification: Use of beneficial micro-organisms such as bacteria or fungi to degrade or bind aflatoxins, reducing their toxic effects.
Physical and Chemical Detoxification	- Physical Methods: Techniques like roasting, steaming, or irradiation to reduce aflatoxin contamination in crops. - Chemical Detoxification: Using agents such as ammonia treatment, ozonation, or activated charcoal to degrade or neutralize aflatoxins in contaminated foods or feed.
Integrated Management Approaches	- Combining Strategies: Use a combination of pre-harvest, post-harvest, biological control, and detoxification methods to minimize aflatoxin contamination at multiple stages of the food production process.

## Data Availability

No new data were created or analyzed in this study. Data sharing is not applicable to this article.

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
