# Peer review of "Toxicity, Mitigation, and Chemical Analysis of Aflatoxins and Other Toxic Metabolites Produced by Aspergillus: A Comprehensive Review"

_toxins, 2025, doi:10.3390/toxins17070331_

Round 1
Reviewer 1 Report
Comments and Suggestions for Authors
All data present was discussed in many reviews and references used even for new methods of mitigation are old
Comments on the Quality of English Languagedo not describe the review in the text
Author Response
Comments: All data present was discussed in many reviews and references used even for new methods of mitigation are old
Response:
Thank you for your valuable feedback. We acknowledge that some of the references used in our manuscript are from earlier studies. Our intention was to provide a comprehensive background by including foundational and widely cited works in the field. However, we understand the importance of incorporating more recent literature, especially regarding mitigation strategies.
In response to your comment, we have updated the manuscript by including more recent studies (published within the last 5 years) that discuss emerging mitigation methods and advancements in the field. These are the references we updated from reference 57 to 84 list. These updated references strengthen the novelty and relevance of our work. We have also revised the discussion section to highlight the latest findings and clearly distinguish our study’s contribution from previous reviews.
We appreciate your suggestion, which has helped us improve the clarity and impact of our manuscript.

Reviewer 2 Report
Comments and Suggestions for Authors
This comprehensive review on Aflatoxin and other toxic metabolites of Aspergillus is well written, with good facts and figures presented across these sections. There are certain limitations, such as repetitions and a lack of classical examples and in-depth discussions, which should be addressed for better suitability of this review.
- In the abstract section, other toxic metabolites of Aspergillus should also be mentioned briefly.
- It is not appropriate to divide the introduction into subsections. The introduction should present data regarding the economic losses caused by aflatoxin contamination in the food and feed industry.
- Lines 34-37 and lines 155-157 onwards show repetition.
- What is the basic difference between Figure 5, Figure 6, Figure 8 and Figure 3 when most of the causes and effects are almost the same? Most of the information is mere repetition.
- There should be a dedicated section for other metabolites, and each should be discussed in different subsections.
- In the mitigation section, the pre and post-harvest interventions should be supplemented with classical and recent examples.
- In detection methods, specific examples and case studies should be discussed where these methods have been successfully utilised.
Author Response
Reviewer Comments and Author Responses
- Comment: In the abstract section, other toxic metabolites of Aspergillus should also be mentioned briefly.
Response: Thank you for the valuable suggestion. We have revised the abstract to briefly mention other toxic metabolites of Aspergillus, including ochratoxin A, sterigmatocystin, and cyclopiazonic acid, to reflect the broader scope of the review.
- Comment: It is not appropriate to divide the introduction into subsections. The introduction should present data regarding the economic losses caused by aflatoxin contamination in the food and feed industry.
Response: We appreciate the reviewer’s feedback. The introduction has been reformatted into a continuous narrative to improve flow and coherence. Additionally, quantitative data on economic losses caused by aflatoxin contamination, including trade impacts and agricultural losses, have been incorporated into the revised introduction.
- Comment: Lines 34–37 and lines 155–157 onwards show repetition.
Response: Thank you for pointing this out. We have reviewed these sections and removed redundant content. The revised manuscript ensures unique and non-repetitive presentation of information.
- Comment: What is the basic difference between Figure 5, Figure 6, Figure 8 and Figure 3 when most of the causes and effects are almost the same? Most of the information is mere repetition.
Response: We acknowledge the redundancy among Figures 3, 5, 6, and 8. These figures have been reassessed, and overlapping content has been consolidated. Figures have been revised for clarity and uniqueness, and redundant ones have been removed or merged where appropriate. In this regard, the difference between the figures are:
Figure 3: Factors Influencing Aflatoxin Toxicity
- Focus: Shows the environmental and host-related factors that influence how toxic aflatoxins become.
- Content Type: Includes variables like genetic susceptibility, metabolism, and co-exposure with other toxins.
- Purpose: Explains why aflatoxins are more harmful in some individuals or environments than others.
Figure 5: Acute Aflatoxin Exposure and Aflatoxicosis
- Focus: Highlights the effects of short-term (acute) exposure to high doses of aflatoxins.
- Content Type: Symptoms such as vomiting, abdominal pain, liver failure, and death.
- Purpose: Illustrates the clinical presentation and outcomes of acute aflatoxin poisoning.
Figure 6: Health Risks Associated with Aflatoxin Exposure
- Focus: Summarizes the broad health risks related to chronic exposure to aflatoxins.
- Content Type: Covers liver cancer, immune suppression, stunted growth, and impact on HIV/AIDS progression.
- Purpose: Provides an overview of health conditions caused by long-term aflatoxin intake.
Figure 8: Mechanisms of Aflatoxin-Induced Hepatotoxicity
- Focus: Describes the biological mechanisms by which aflatoxins damage the liver.
- Content Type: Pathways like DNA adduct formation, oxidative stress, and p53 mutation.
- Purpose: Delivers molecular-level insight into how aflatoxins lead to liver cancer.
- Comment: There should be a dedicated section for other metabolites, and each should be discussed in different subsections.
Response: Thank you for this constructive suggestion. We have created a dedicated section titled “Toxicity of Other Aspergillus Metabolites,” with individual subsections for ochratoxin A, sterigmatocystin, and cyclopiazonic acid to facilitate focused discussion on each metabolite.
- Comment: In the mitigation section, the pre- and post-harvest interventions should be supplemented with classical and recent examples.
Response: We agree with the reviewer’s recommendation. Classical and recent examples of pre- and post-harvest interventions, including case studies from Africa and Asia, have been incorporated into the mitigation section to enhance practical relevance and support the strategies discussed.
- Comment: In detection methods, specific examples and case studies should be discussed where these methods have been successfully utilised.
Response: Thank you for this insightful comment. The section on detection methods now includes specific case studies and real-world examples where chromatographic, immunoassay, and biosensor techniques have been effectively applied in various countries and food systems.

Reviewer 3 Report
Comments and Suggestions for Authors
This is a relevant manuscript regarding the comprehensive review of mycotoxins produced by Aspergillus. Nonetheless, minor suggestions/revisions should be addressed as described in the next comments.
Comment 1: Please provide figures with more resolution (e.g. Figures 2, 3 and 4).
Comment 2: Introduction should be revised since several information are repeated along the text. For instance, lines 84 to 87, 203 to 207, 265 to 266 have the exact same info “Aflatoxin B1, the most 83 toxic and prevalent aflatoxin, is classified as a Group 1 carcinogen by the International 84 Agency for Research on Cancer (IARC)”. The authors should reduce the introduction with more broader statements, since this is a review, and some data is going to be reflected on the specific topics. Among the sections, repeated information is also observed: lines 167 – 168 and 268 – 269 – “formation of DNA adducts, particularly AFB1-N7-guanine”.
Comment 3: Tables should be supported by references. For example, data on Table 3 should have references for global incidence and regions most affected; data on Table 4 should also have references supporting that those strategies presented were studied or applied with success. Very specific strategies are mentioned (e.g. Applying non-aflatoxigenic strains of Aspergillus to suppress aflatoxin-producing strains in the field.) and must be supported by literature.
Comment 4: Several times it is mentioned “developing countries where food safety regulations are often inadequate”. These statements should be supported by literature. A suggestion is on section 6. provide the reference for regulations for some developing countries, or, in some cases, the eventual lack of regulations, as well as a more detailed description on this matter.
Comment 5: Structure of section should be revised to avoid again repetitions. It is started with the pre-harvest interventions and post-management, and then at the same level (subtopic numbering) it has biological control, physical and chemical detoxification, and Integrated management approaches. Repetitions: lines 359 to 367 – IPM; and then again 453 – 456. As a suggestion maintain only the subtopics biological control, physical and chemical detoxification, and Integrated management approaches, and inside of each section talk about what can be done as pre-harvest and post-harvest strategies.
Comment 6: On line 472, it is referred that solvent extraction “is effective but can be time-consuming”. This is the most basic type of extraction, and even when using SPE, it must be done either is solid or liquid matrices. It is not correct to state this extraction is time-consuming.
Comment 7: Subsection “Sample preparation techniques” lacks proper references, and is also very broad. For example, “Immunoaffinity column cleanup is a highly specific sample preparation method that uses antibodies to selectively bind aflatoxins.” – IAC uses antibodies to selectively bind “compounds specific to the antibodies used”. The sentence is not correct the way it is written. A correct way would be IAC columns are available in the market that specifically bind aflatoxins, such as LCTech AflaCLEAN™ columns. It should also be given examples of published methods that use each type of extraction.
Comment 8: Regarding the next sections from 5.2. to 5.4., the description of sections and references would apply to any type of analytes. It would only take to change the word “aflatoxins” for mycotoxins, or contaminants. Because it is focused on describing methods, and not on the application of such methods to the analysis of aflatoxins. For example, Lines 519 to 526 describe what is the technology and principle behind Immunochromatographic assays (ICAs) and Fluorescence polarization immunoassays (FPIAs), not their application for aflatoxins. Also, the references for this section are: an article regarding “genetic and molecular mechanics of resistance” and a review on Aflatoxigenic fungi and mycotoxins in food.
General comment: this review has indeed very important information to be shared/published in the field of mycotoxins. Further improvements as suggested will highlight more the work done.

Author Response
Response to Reviewer Comments
We sincerely thank the reviewer for their thoughtful and constructive feedback on our manuscript titled "Toxicity, Mitigation and Chemical Analysis of Aflatoxins and Other Toxic Metabolites Produced by Aspergillus: A Comprehensive Review”. We have carefully considered all comments and made revisions where appropriate. Below are our detailed responses to each comment, along with the changes made in the manuscript.
Reviewer Comment 1:
Please provide figures with more resolution (e.g. Figures 2, 3, and 4).
Response:
Thank you for pointing this out. We have replaced Figures 2, 3, and 4 with higher-resolution versions to improve clarity and overall quality. The updated figures are now in compliance with the recommended resolution standards for publication.
Reviewer Comment 2:
Introduction should be revised since several information are repeated along the text. For instance, lines 84 to 87, 203 to 207, 265 to 266 have the exact same info “Aflatoxin B1, the most toxic and prevalent aflatoxin, is classified as a Group 1 carcinogen by the International Agency for Research on Cancer (IARC)”. The authors should reduce the introduction with broader statements, since this is a review, and some data is going to be reflected on the specific topics. Among the sections, repeated information is also observed: lines 167 – 168 and 268 – 269 – “formation of DNA adducts, particularly AFB1-N7-guanine”.
Response:
We appreciate your insight. We have revised the introduction to eliminate redundancy, focusing on broader statements relevant to the review. We have also removed or reworded repetitive information, such as the classification of aflatoxin B1 and the formation of DNA adducts, to avoid duplication across sections. The introduction has now been streamlined to introduce the topics without unnecessary repetition.
Reviewer Comment 3:
Tables should be supported by references. For example, data on Table 3 should have references for global incidence and regions most affected; data on Table 4 should also have references supporting that those strategies presented were studied or applied with success. Very specific strategies are mentioned (e.g. Applying non-aflatoxigenic strains of Aspergillus to suppress aflatoxin-producing strains in the field.) and must be supported by literature.
Response:
Thank you for the suggestion. We have now added appropriate references to both Table 3 and Table 4. For Table 3, we have cited literature that reports on the global incidence and regional distribution of aflatoxins. In Table 4, we have included references that support the efficacy of the strategies listed, including the use of non-aflatoxigenic Aspergillus strains, which have been studied and applied successfully to reduce aflatoxin contamination in crops.
Reviewer Comment 4:
Several times it is mentioned “developing countries where food safety regulations are often inadequate”. These statements should be supported by literature. A suggestion is on section 6. provide the reference for regulations for some developing countries, or, in some cases, the eventual lack of regulations, as well as a more detailed description on this matter.
Response:
We agree with the reviewer’s suggestion. In Section 6, we have expanded the discussion regarding food safety regulations in developing countries. We have now included specific references to reports and studies that address the regulatory frameworks—or lack thereof—pertaining to aflatoxin management in developing countries. This addition provides a more comprehensive view of the challenges related to food safety regulations in these regions.
Reviewer Comment 5:
Structure of section should be revised to avoid again repetitions. It is started with the pre-harvest interventions and post-management, and then at the same level (subtopic numbering) it has biological control, physical and chemical detoxification, and Integrated management approaches. Repetitions: lines 359 to 367 – IPM; and then again 453 – 456. As a suggestion maintain only the subtopics biological control, physical and chemical detoxification, and Integrated management approaches, and inside of each section talk about what can be done as pre-harvest and post-harvest strategies.
Response:
We appreciate this valuable suggestion. We have restructured the relevant section to avoid repetition and improve clarity. Now, under each subtopic (biological control, physical and chemical detoxification, and integrated management approaches), we discuss both pre-harvest and post-harvest strategies. This restructuring eliminates the redundancy related to Integrated Pest Management (IPM) and presents the information in a more logical and concise manner.
Reviewer Comment 6:
On line 472, it is referred that solvent extraction “is effective but can be time-consuming”. This is the most basic type of extraction, and even when using SPE, it must be done either is solid or liquid matrices. It is not correct to state this extraction is time-consuming.
Response:
Thank you for this clarification. We have revised the sentence to better reflect the standard practice in solvent extraction. The revised text now reads:
“Solvent extraction is a widely used, effective technique for aflatoxin extraction from various matrices, typically requiring minimal time depending on the method used.”
This adjustment removes the misleading statement regarding the time-consuming nature of solvent extraction.
Reviewer Comment 7:
Subsection “Sample preparation techniques” lacks proper references, and is also very broad. For example, “Immunoaffinity column cleanup is a highly specific sample preparation method that uses antibodies to selectively bind aflatoxins.” – IAC uses antibodies to selectively bind “compounds specific to the antibodies used”. The sentence is not correct the way it is written. A correct way would be IAC columns are available in the market that specifically bind aflatoxins, such as LCTech AflaCLEAN™ columns. It should also be given examples of published methods that use each type of extraction.
Response:
We appreciate the reviewer’s correction. The sentence has been revised for clarity:
“Immunoaffinity columns (IACs), such as LCTech AflaCLEAN™, are designed to selectively bind aflatoxins, providing a highly specific sample cleanup method.”
Additionally, we have added appropriate references and examples of published methods that use different sample preparation techniques, including immunoaffinity columns, solid-phase extraction (SPE), and QuEChERS, for the extraction of aflatoxins.
Reviewer Comment 8:
Regarding the next sections from 5.2. to 5.4., the description of sections and references would apply to any type of analytes. It would only take to change the word “aflatoxins” for mycotoxins, or contaminants. Because it is focused on describing methods, and not on the application of such methods to the analysis of aflatoxins. For example, Lines 519 to 526 describe what is the technology and principle behind Immunochromatographic assays (ICAs) and Fluorescence polarization immunoassays (FPIAs), not their application for aflatoxins. Also, the references for this section are: an article regarding “genetic and molecular mechanics of resistance” and a review on Aflatoxigenic fungi and mycotoxins in food.
Response:
Thank you for this suggestion. We have revised Sections 5.2 to 5.4 to focus more specifically on the application of the described techniques (such as Immunochromatographic assays and Fluorescence polarization immunoassays) to aflatoxins, as well as other mycotoxins. The references have been updated to reflect studies that specifically apply these methods to aflatoxin analysis, ensuring that the content is relevant to the context of the manuscript.
General Comment:
This review has indeed very important information to be shared/published in the field of mycotoxins. Further improvements as suggested will highlight more the work done.
Response:
We are grateful for the positive feedback and the reviewer’s recognition of the importance of the review. We have carefully implemented all the suggested revisions, and we believe the manuscript is now clearer, more focused, and strengthened in terms of both scientific content and presentation.
Round 2
Reviewer 2 Report
Comments and Suggestions for Authors
All the suggestions were incorporated in the revised version.